# Associations between Lifestyle Changes, Risk Perception and Anxiety during COVID-19 Lockdowns: A Case Study in Xi’an

**DOI:** 10.3390/ijerph192013379

**Published:** 2022-10-17

**Authors:** Huan Yang, Qingyun Zhao, Zhengkai Zhang, Wenxiao Jia

**Affiliations:** College of Landscape Architecture and Art, Northwest A&F University, Yangling 712100, China

**Keywords:** lifestyle changes, anxiety, risk perception, COVID-19, Xi’an

## Abstract

The outbreak of COVID-19 dramatically changed individuals’ lifestyles, which in turn triggered psychological stress and anxiety. Many previous studies have discussed the relationships between lifestyle changes and anxiety and risk perception and anxiety independently. However, few papers have discussed these factors in a comprehensive and systematic manner. We established a six-dimensional system to assess changes in individuals’ lifestyles, which include dietary habits, physical activity (PA), sleep, screen time, smoking and alcohol consumption, and interaction with neighbors. Then, we collected information relating to socio-demographics, lifestyle changes, risk perception, and anxiety, and discussed their associations using multilinear and stepwise logistic regressions. The results show that not all lifestyle changes had an influence on anxiety. Changes in PA and interaction with neighbors were not significantly associated with anxiety. Risk perception was found to be inversely related to anxiety. Changes in dietary habits, family harmony, and net income were negatively related to anxiety among the group with higher risk perception. As individuals perceived a higher severity of COVID-19, the impact of their financial status on anxiety increased. These findings provide a valuable resource for local governments seeking to refine their pandemic strategies by including approaches such as advocating healthy lifestyles and stabilizing the job market to improve individuals’ mental health during lockdowns.

## 1. Introduction

As of 8 August 2022, there had been 581 million confirmed cases of COVID-19 and almost 6 million deaths globally, as reported by the WHO [1]. Furthermore, the global economy faced a dramatic slowdown as a result of pandemic prevention and control measures. According to a study in New Zealand, the unemployment rate doubled from 5.2% before to 10.5% by the third week of lockdowns in 2020. Approximately 44% of individuals lived in a household in which members experienced job and/or income loss [2]. You et al. (2020) estimated that the total monthly economic losses in Wuhan caused by lockdowns reached CNY 177.0413 billion [3]. This evidence indicates that this global health emergency has not only caused physical harm and economic loss, but also had an unprecedented influence on individuals’ mental well-being. A series of measures, such as containment, quarantine, community controls, transitioning to online services, and school closures were adopted to prevent and control the COVID-19 pandemic, which had a significant impact on individuals’ daily lives. Such changes can substantially affect mental health. Sixty percent of the participants in a global cross-sectional survey reported anxiety symptoms during the pandemic [4]. According to Li Shi (2020), approximately one-quarter to one-third of respondents exhibited symptoms of depression, anxiety, insomnia, and acute stress during the COVID-19 pandemic in China [5], and thus the prevalence of mental health symptoms during was higher than it was before the outbreak of the pandemic.

Various studies have discussed the associations between lifestyle changes and anxiety. Currently, lifestyle is seen as a multidimensional construct that encompasses diet, physical activity, sleep, screen time, smoking and alcohol consumption, stress management, social support, and digital technology usage. However, previous studies assessed specific lifestyle behaviors and did not take a comprehensive approach to the causes of anxiety [6,7,8,9]. McDowell et al. (2019) and Maher et al. (2021) only focused on physical activity (PA) and its associations with anxiety [10,11]. Stanton et al. (2020) investigated changes in PA, sleep, and tobacco and alcohol consumption and their associations with anxiety in Australian adults during the COVID-19 pandemic [12]. Grey et al. (2020) examined the role of social support and found it to be inversely related to anxiety during quarantine [13]. In China, the existing research has paid more attention to the three traditional pillars of lifestyle: diet, PA, and sleep. Compared with the physical activity of Chinese individuals during the non-pandemic period (14.1%) reported by WHO, the prevalence of insufficient physical activity rose over three-fold (57.5%) during the pandemic [14]. Zheng et al. (2022) discussed the impacts of the COVID-19 pandemic on diet among Chinese citizens [15]. Zhu et al. (2021) found a significant increase in total food intake (9.8%) and a significant decrease in PA (31.5%) during lockdowns in Wuhan [16]. Song et al. (2022) confirmed that anxiety was associated with problematic smartphone use and sleep disturbance among medical students during the COVID-19 pandemic in Shen Yang, China [17]. Few studies have focused on other aspects of lifestyle changes, such as smoking and alcohol consumption and social support during the pandemic in China, and the studies on the associations between multidimensional lifestyle changes and anxiety are insufficient. Thus, this paper seeks to explore the role of lifestyle changes from multiple dimensions following the three traditional pillars of lifestyle: nutrition, physical activity, and restorative sleep [18]. Moreover, the lifestyle factors discussed in this study include screen time, smoking and alcohol consumption, and social support, which are considered important components of individuals’ daily lives. Social support is derived from three sources: family, friends, and other [19]. In China, it is said that “a close neighbor is better than a distant relative”. Additionally, lockdowns had a significant impact on neighbor interaction, as individuals were prevented from communicating with their neighbors in person. Thus, this study includes six dimensions of lifestyle changes, which encompass dietary habits, PA, sleep, screen time, smoking and alcohol consumption, and interaction with neighbors.

Risk perception is the cognitive response and subjective judgment that compares the probability that an event will occur with the seriousness of its potential damage [20,21]. With regard to the connotation of risk perception, some studies have inquired into worries and concerns, others have assessed fear or nervousness, and still others have measured the likelihood of certain events [22]. Raaijmakers et al. (2008) took awareness, worry, and preparedness as the three characteristics of the risk of natural disasters [23]. In terms of the COVID-19 pandemic, some studies have assessed the risks of contracting or dying from it, being quarantined, losing their job (if currently working), and running out of money [24]. Lanciano et al. (2020) built a multidimensional framework to explore risk perceptions in terms of health, mortality, work, and economic, interpersonal, and psychological risks [22]. Chen et al. (2021) categorized risk perception according to perceived social risk and the perceived risk of being infected [25]. Various studies have found that higher risk perceptions were associated with greater anxiety [26,27,28]. Malesza et al. (2021) found that a greater perceived risk of infection and likelihood of contracting COVID-19 led to greater anxiety among individuals [27]. However, few studies have discussed the significance of the factors that influence anxiety among groups with different risk perceptions.

In general, lifestyle changes and risk perception both have an obvious influence on anxiety. Previous articles have mainly focused on the three traditional pillars of lifestyle changes and their associations with anxiety. In addition, few studies have discussed the associations between lifestyle changes, risk perception, and anxiety in the context of the COVID-19 pandemic, especially in China. Therefore, this paper exploring the relationship between individuals’ lifestyle changes, risk perception, and anxiety during the COVID-19 pandemic had theoretical implications. In addition, the findings of this paper would provide references for individuals and local governments to maintain a healthy lifestyle, weaken risk perception, and ease anxiety symptoms.

The aims of this paper are (1) to explore the associations between lifestyle changes, risk perception and anxiety, and (2) to find the significant factors that influence anxiety among groups with different levels of risk perception.

## 2. Materials and Methods

### 2.1. Study Areas

As the capital city of Shaanxi province, Xi’an is the ninth-largest Chinese city, with a population of about 13 million, and both a national and international transportation hub city. During the 40 days of the Spring Festival travel rush in 2022, 7.8 million passengers were delivered by the Xi’an railway system. Since the first reported case in Xi’an on 9 December 2021, the number of new cases per day reached a peak of 174 on 31 December 2021, about one month before the Chinese New Year. With the coming Spring Festival travel rush and the rapid spread of the COVID-19 Delta variant, Xi’an was caught in a very urgent situation. In response, it declared a lockdown on 23 December 2021. During that period, the government implemented compulsory measures such as containment, quarantine, community controls, a transition to online business, and school closures. Residents were obliged to stay at home unless they were able to take nucleic acid tests. Most residents did not need to go to the market to buy daily goods, which were supplied through unified community distribution to reduce infection via social contact. These measures proved to be effective, and Xi’an declared that it was open on 24 January 2022, one week before the Spring Festival. From outbreak to openness, Xi’an took one month to achieve zero daily new cases. The compulsory measures adopted in Xi’an during the lockdown period were similar to those in other cities. Thus, choosing Xi’an as a case study through which to explore the associations between lifestyle changes, risk perception, and anxiety is justified.

### 2.2. Questionnaire Design

The questionnaire, which was written in Chinese, consisted of five parts: individual and family information, lifestyle changes, risk perception, and anxiety status. The individual information included gender, age, occupation, net income, living status, work status (if they needed to work from home during lockdowns), and having special care groups (e.g., children under 6 years old, pregnant women, or chronic patients). Family information was collected by asking “How harmonious would you rate your family?” and “How crowded would you rate your house?” The questions were answered on a 5-point Likert scale.

Lifestyle changes encompassed dietary habits, physical activity, sleep, screen time, smoking and alcohol consumption, and interaction with neighbors. Changes in lifestyle behaviors were assessed using questions such as: “How did your dietary habits/physical activity/sleep/screen time/smoking and alcohol consumption habits/interaction with neighbors change during the lockdown period as compared to the pre-pandemic period?” with three possible answers: “became less healthy/frequently”, “no change” and “became healthier/frequently”. This paper assessed individuals’ risk perception of COVID-19 in Xi’an with the following question: “How serious was the COVID-19 pandemic in Xi’an compared to in Wuhan in 2019?” with the three choices of: “less serious than Wuhan (Lower Group)”, “as serious as Wuhan (Medium Group)”, and “more serious than Wuhan (Higher Group)”.

Anxiety status was measured using the Generalized Anxiety Disorder-7 (GAD-7), which is an efficient tool that is widely accepted in clinical practice and research. Participants were asked to answer “Over the last 2 weeks, how often have you been bothered by the following seven problems?” with responses including “Feeling nervous, anxious or on edge” and “Having trouble relaxing”. The four options indicate the degree to which participants agreed with each item ranked from 0–3, meaning “Not at all”, “Several days”, “More than half of the days”, and “Nearly every day”, in that order. The total score ranged from 0 to 21. Anxiety states were divided into four categories: normal (0–4), mild (5–9), moderate (10–14), and severe (15–21) [29].

### 2.3. Data Collection

An anonymous online survey was published via Survey Star (Changsha Ranxing Science and Technology, Shanghai, China), a widely used questionnaire distribution platform in China, from 6 January 12 January 2022 (i.e., two weeks after Xi’an declared lockdown). The authors of this study were local residents of Xi’an. First, the link and QR code of the online questionnaire were delivered to friends, family members, colleagues, students, and working partners of authors via WeChat (Tencent, Shenzhen, China) using the snowball sampling method. Then, the link and QR code of the online questionnaire were also posted in their “community discussion group”. This group was formed spontaneously among community residents and played an important role in information exchange during the lockdown period. The survey took about 10 min to complete and could only be completed once.

### 2.4. Statistical Analysis

The descriptive statistics, including frequencies and percentages, were generated for categorical variables, such as gender, work status, family harmony, lifestyle changes, and risk perception. Means and standard deviations (SDs) were generated for anxiety scores. A non-parametric test was used to evaluate the statistically significant differences in anxiety status based on participants’ sociodemographic information, lifestyle changes, and risk perception. Multilinear regression was used to explore the associations between lifestyle changes and anxiety. The changes in dietary habits, PA, sleep, screen time, smoking and alcohol consumption, and interaction with neighbors were independent variables, while the total anxiety score was the dependent variable. Then, this paper used stepwise logistic regression to explore the relationships between lifestyle changes, risk perception, and anxiety. Model 1 took lifestyle changes as independent variables. Model 2 took lifestyle and risk perception as independent variables. Crude estimates using Model 1 and Model 2 were reported with 95% confidence intervals (CIs). Later, the research categorized samples by risk perception and formed three groups (i.e., Lower, Medium, and Higher). Multilinear regression was adopted to explore the statistically significant factors that affect anxiety among these three groups. The independent variable was the total anxiety score. Dependent variables included lifestyle changes as well as individual and family information. All statistical analyses were run on an IBM SPSS 26.0 (IBM, Armonk, NY, USA). All *p*-values were two-sided and considered significant if less than 0.05.

## 3. Results

### 3.1. Individual and Family Characteristics

A total of 1331 questionaries were received. After eliminating incomplete and unusable questionnaires, the final sample consisted of 1272 valid responses for an effective response rate of 95.57%. The samples had balanced gender distribution: 54% females and 46% males. Overall, 79.6% of participants’ net income was only sufficient to meet daily basic expenses. It was found that 87.70% of the respondents lived with others. The number of people who needed to work at home (57.10%) was slightly higher than those who did not (42.90%). The questionnaires found that 70.8% of the participants were between 19 and 45 years of age. Families with special members comprised 41.2% of the sample. More than half of the participants (63.2%) perceived that their family was very harmonious and that their indoor space was not crowded (62.7%). Relatively few respondents (2.7%) perceived that their family was not harmonious and that their indoor space was very crowded (8%) (Table 1).

### 3.2. Characteristics of Lifestyle Changes, Risk Perception, and Anxiety during the Xi’an Lockdown Period

PA changed most significantly (77.2%) compared to the pre-pandemic period, followed by screen time (77.1%), interaction with neighbors (68.2%), sleep (62.9%), dietary habits (48.5%), and smoking and alcohol consumption (13.9%). Negative changes in screen time (71.6%) ranked first, followed by PA (69.7%), interaction with neighbors (54.9%), sleep (48.9%), dietary habits (22.2%), and smoking and alcohol consumption (7.8%). Overall, 75.3% of respondents perceived that the severity of COVID-19 in Xi’an at the end of 2021 was not higher than it was in Wuhan at the end of 2019. Additionally, 63.8% of respondents reported minimal anxiety and 24.2% had mild anxiety, while 12% of participants had moderate-to-severe anxiety. Individuals’ anxiety had statistically significant differences among individual and family characteristics, lifestyle changes, and risk perceptions, except gender and work status, using a non-parametric test (Table 1).

### 3.3. Associations between Lifestyle Changes, Risk Perception and Anxiety

This paper used multilinear regression to explore the associations between lifestyle changes and anxiety. Changes in dietary habits, physical activity, sleep, screen time, smoking and alcohol consumption, and interaction with neighbors were included as independent variables, and anxiety was the dependent variable after controlling for individual and family characteristics. Except for the changes in interaction with neighbors and PA, the other aspects of lifestyle changes were negatively associated with anxiety (Table 2).

Stepwise logistic regression was used to explore the relationships between lifestyle changes, risk perception, and anxiety. The unadjusted Model 1 took lifestyle changes as independent variables. Model 2 adjusted for risk perceptions. Negative changes in screen time and risk perception were significant factors for participants with mild anxiety. For individuals with moderate-to-severe anxiety and risk perception, negative changes in dietary habits and sleep were statistically significant influencing factors. Adjusting risk perception did not significantly affect the relationships between lifestyle changes and anxiety, except for the dimension of smoking and alcohol consumption. Among respondents with moderate-to-severe anxiety status, negative changes in smoking and alcohol consumption were significant in Model 1, but not in Model 2. In addition, those who reported negative changes in screen time habits were more likely to suffer mild anxiety (OR: 1.60). Individuals with negative changes in dietary habits and sleep were more likely to suffer from moderate-to-severe anxiety. The ratios were 3.94 and 1.70, respectively. Table 3 shows that the respondents with lower risk perception were 0.69 times as likely to suffer from mild anxiety as those with general risk perception. This ratio in the group with moderate-to-severe anxiety was 0.58.

### 3.4. Influencing Factors of Anxiety among Groups with Different Levels of Risk Perception

The results of the multilinear regression showed that changes in dietary habits and family harmony were negatively associated with anxiety, no matter the level of risk perception. As for the group that perceived the severity of COVID-19 in Xi’an at the end of 2021 as being lower than it in Wuhan at the end of 2019, changes in dietary habits, sleep, family harmony, and indoor space were the factors that most significantly influenced anxiety. In the Medium Group, the noticeable variables were “work status” and “having special family members to care for”. Participants who needed to keep working during the lockdown period had a lower level of anxiety. As for individuals whose family had special care members, their level of anxiety was higher than those who did not. In the Higher Group, net income was significantly negatively associated with anxiety in addition to changes in dietary habits and family harmony (Table 4).

## 4. Discussion

### 4.1. Characterisitcs of Lifestyle Changes and Anxiety

The observed lifestyle changes during lockdowns in Xi’an at the end of 2021 were almost consistent with those in previous studies. It was confirmed that screen time and physical activity were the lifestyle behaviors most sensitive to the effects of strict lockdowns and mandatory home isolation [30,31]. A study in Spain showed that most participants reported substantial changes in outdoor time (93.6%) and physical activity (70.2%) during the COVID-19 pandemic [6]. He et al. (2020) showed that the average steps per day and exercise declined significantly for both males and females during the semi-lockdowns, which resulted in weight gain in China [32]. In addition to these findings, a noticeable result should be mentioned. Changes in dietary habits showed a slight positive trend as a whole. The proportion of positive changes in dietary habits was 26.3%, while the proportion of negative changes was 22.2%. The reason for this slight positive trend seemed to be that participants had more time to cook at home during lockdowns. López-Moreno et al. found that 73.5% of participants reported increases in cooking at home during lockdowns in a Spanish population [33]. Similarly, Ruiz-Roso et al. (2020) found that individuals had more time to cook at home and a heathier food intake in adolescents. Individuals’ dietary habits became much healthier in general. Vegetables and fruit intake significantly increased, while fast food intake was dramatically reduced during the pandemic [34].

Eighty-eight percent of the participants in Xi’an presented with minimal or mild anxiety symptoms. This was inconsistent with previous studies. Liu et al. (2021) found that 51% of the participants presented with mild, moderate, or severe anxiety symptoms during the first COVID-19 outbreak in Wuhan [35], while the proportion in Xi’an was 36.2%. Wang et al. (2020) found that 28.8% of respondents reported moderate or severe anxiety symptoms during the initial stage of the COVID-19 pandemic in China. This was almost two times higher than it was in Xi’an (12%) [36]. On the whole, the level of anxiety during the lockdown period in Xi’an at the end of 2021 was lower than it was in Wuhan during the lockdown period at the end of 2019. One of the potential reasons behind this phenomenon was that people became accustomed to COVID-19 and home isolation. Since the first outbreak in Wuhan, the pandemic spread was sporadic over the past two years and the control measures had been normalized. Citizens of Xi’an were required to undergo nucleic acid testing twice per month. Individuals who entered crowded places, such as subways, buses, or shopping malls, were required to check their health status through an app on WeChat. The ongoing outbreak and normalized control conditions enhanced individuals’ resilience to COVID-19 and attenuated their fear and negative emotions when facing the return of the pandemic. In addition, some scholars found public emotions or behaviors were associated with content on social media [37,38]. Zhao et al. (2020) found the public attention given to the COVID-19 epidemic can be divided into three stages. In the first stage, most of the topics related to the COVID-19 epidemic were about the notification of the virus. During this stage, more hot research topics mentioned information about prevention or protection. The public reduced their previous levels of worries and fears about the epidemic. Their positive emotions increased [37]. Social media acts as a double-edged sword. These platforms may spread fake messages and instigate panic amongst members of the general public. Thus, objective and scientific information about COVID-19 is important.

### 4.2. Associations between Lifestyle Changes and Anxiety

The multilinear regression showed that changes in diet, sleep, screen-time habits and smoking and alcohol consumption negatively influenced anxiety. These conclusions were consistent with some previous studies [4,8]. For example, some studies noted that unhealthy dietary habits would aggravate negative emotional states and increase the risk of suffering from anxiety [39,40]. Consumption of poor diets had been shown to promote nitric oxide production, which facilitates anxiety in humans [41]. In addition to these findings, this paper also found some unexpected results which were not in line with previous studies.

First, changes in PA had no significant influence on anxiety. This was inconsistent with the finding of Stanton et al. (2020), in which anxiety was significantly associated with changes in PA both independently and as a composite score in Australian adults [12]. The possible reason for this result lies in the influences from other lifestyle considerations, especially changes in screen time. Lu et al. (2020) indicated that PA had a weak association with anxiety symptoms, and a high level of PA may not alleviate the effect of excessive sitting on mental health outcomes [42]. Trinh et al. (2015) found that PA had a lower effect on anxiety than screen time [43]. Second, this paper found changes in interaction with neighbors were not significantly related with anxiety. The reason for this result may be related to positive changes in family support during lockdowns. According to the “Bulletin of the 2018 National Time Use Survey” from the National Bureau of Statistics of China, the time individuals spent caring for adult family members was 2 h and 43 min. Communications between family members were limited. However, individuals had more time and opportunities to accompany family members during lockdowns. The positive changes in family ties relieved individuals’ anxiety and weakened the influence of interaction with neighbors on anxiety. This finding was confirmed by Ozmete et al. (2020) and Xu et al. (2020). Ozmete et al. (2020) found that individuals’ perceived family and friend support were higher than other types of support during the lockdown period [19]. According to Xu et al. (2020), only perceived social support from family buffered the detrimental effects from trait loneliness due to COVID-19 anxiety during the peak of the pandemic, whereas perceived social support from friends and significant others did not moderate this relationship [44]. The other possible explanation for the association between anxiety and interaction with neighbors was the usage of social media during the lockdown period. Various papers have showed that social media use is an emotional resource for social support in times of crisis or isolation [45]. Social media platforms were accessible to every individual, allowing them to share, post, and react to any medical information regarding the pandemic [46]. Although neighbors could not interact in person, they could receive support from neighbors via social media platforms such as WeChat.

### 4.3. Comprehensive Associations between Lifestyle Changes, Risk Perception, and Anxiety

The result of the stepwise logistic regression showed that risk perception was negatively associated with anxiety. The lower risk severity individuals perceived, the lower the possibility the individual suffered anxiety. Respondents with lower risk perception were less likely to have anxiety (mild, OR: 0.69; moderate-to-severe, OR: 0.58). This finding was consistent with those of previous studies [26,27]. According to the psychometric paradigm, the emergence of sudden-onset disaster often arouses “dread” and “risk of the unknown” [47], which induce stress and negative emotions and in turn affects mental health [26,48,49,50]. Individuals’ risk perception had no effect on the association between lifestyle changes and anxiety, except the negative changes in smoking and alcohol consumption. This suggests that most lifestyle changes and risk perceptions were significant factors in anxiety.

In addition, the variables that affect anxiety differed as individuals’ risk perception varied. In the Medium Group, work status and having special family members to care for were noticeable factors that were significantly associated with anxiety. It is worth noting that individuals who needed to keep working during lockdowns were less likely to suffer from anxiety symptoms. This finding was confirmed by the study of Sultana et al. (2020), in which participants who did not receive salary income during the pandemic were more likely to suffer from anxiety symptoms [51]. Individuals who had special family members to care for during the lockdown period were more likely to suffer from anxiety. Similarly, some previous studies reported that having elderly people and children in the family increases the risk of mental health problems [52]. In the Higher Group, net income was inversely associated with anxiety. The spread of the COVID-19 pandemic brought a slowdown in economic growth. Some small businesses closed or laid off workers. When individuals perceived higher risk severity, they were more likely to worry about losing their job or income to cover their mortgage. Lanciano et al. (2020) found individuals perceived the economic risks (e.g., unemployment, managing work, job prospects, etc.) as being highest during the pandemic [22]. According to the 2021 Xi’an statistical bulletin, the annual per capita disposable income was RMB 38,701 (Xi’an People’s Government, 2021), while the average price of commercial housing was above 15,000 RMB per square meter (Fang tianxia, 2022). In Xi’an, most families had mortgages and car loans to cover each month. The heavy burden and perceived risk of unemployment were potential triggers for anxiety. This was consistent with a systematic review and meta-analysis by Richardson et al. (2013), in which debt increased the odds of neurotic disorders such as anxiety over threefold (OR = 3.21). Mortgage debt and financial hardship have been found to have strong ties to mental illness [53,54]. The evidence in Wuhan also confirmed economic loss caused by the pandemic as a major source of psychological distress [55]. On the whole, as individuals perceived higher risk severity of COVID-19, the impact of financial status on anxiety was found to be greater. In addition, family harmony was negatively associated with anxiety regardless of the level of individuals’ risk perception. This was in line with the finding that interaction with neighbors had no significant effect on anxiety. Perceived support from family, rather than from neighbors, could therefore relieve anxiety symptoms.

To the best of our knowledge, this was the first published account of associations between lifestyle changes, risk perception and anxiety, especially in the context of China. It explored the influencing factors of anxiety among groups with different levels of risk perception. Additionally, it established a six-dimensional lifestyle framework and discussed changes in dietary habits, PA, sleep, screen time, smoking and drinking, and interaction with neighbors during lockdowns in Xi’an at the end of 2021. However, there were some limitations that must be considered. First, the participants may have a selection bias. The snowball method may lead to the sample being non-probabilistic in nature. The survey conducted via Survey Star excluded individuals who did not regularly use the Internet, such as seniors older than 65 and children younger than 10 years of age. Some previous studies found younger age groups had a higher risk of anxiety and were more vulnerable to anxiety symptoms [56]. Older individuals (i.e., 55 years old and above) reported higher resilient coping and lower anxiety scores [4]. Thus, the results of the online questionnaire survey could be trusted. Additionally, the anxiety of children and elderly groups could be explored in future research. Second, it has been nearly two years since the outbreak of the pandemic. During this period, individuals’ resilience to COVID-19 control measures may have changed. This research was a cross-sectional study. The continuous impact of COVID-19 lockdown measures on individuals’ lifestyle changes, risk perception, and anxiety require longitudinal research in the future. Third, social support has a rather broad connotation. This paper only considered the dimension of interaction with neighbors. The same situation occurred in terms of risk perception, which was only one dimension of anxiety. As such, a more systematic analysis should be conducted in the future.

## 5. Conclusions

Our findings provide a reference for individuals and local governments. Local governments should not only pay attention to economic recovery and individuals’ physical health, but should also focus on their mental health during lockdowns to create more efficient policies in this regard. First, healthy lifestyles were found to be helpful for relieving anxiety. Positive changes in diet, sleep, screen time habits, and smoking and alcohol consumption should be advocated by local governments or communities. During lockdown periods, individuals should reduce screen time and avoid staying up late. Additionally, individuals should enjoy the time they spend with family members, partake in indoor fitness activities every day, keep a peaceful mind, improve their resilience to stress or anxiety, and create a harmonious family atmosphere. Second, individuals’ risk perception was inversely associated with anxiety. Disseminating accurate and scientific knowledge about emerging infectious diseases to ease public risk perception is therefore necessary. Individuals should follow official media platforms. Policymakers should intervene to reduce unnecessary risk perceptions or avoid excessive concern to mitigate negative emotions [49]. Third, the results of this paper showed that the significant factors of anxiety among groups with different levels of risk perception were not the same. Specific strategies should therefore be adopted for different groups. Local governments should pay particular attention to certain groups, such as low-income, unemployed, or special care groups. Financial policies should be adopted to stabilize the job market and decrease the risk of economic loss, such as tax cuts and unemployment subsidies. Additionally, it is essential to ensure accessibility to hospitals during lockdowns by providing fast-track access or establishing designated hospitals. In addition, family harmony was negatively associated with anxiety regardless of the level of risk perception. During lockdown periods, individuals can share and communicate with family members to bring each other closer together and enhance family cohesion to ease individuals’ anxiety.

## Figures and Tables

**Table 1 ijerph-19-13379-t001:** Descriptive analysis and non-parametric test of samples.

	Categories		Percentage (%)/(Mean + SD)	*p*-Value
Individual and family characteristics	Gender	Male	46.4	0.745
		Female	53.6	
	Net income	Make ends meet	40.30	0.000 *
		Flat	39.30	
		Slight surplus	12.30	
		Enough	8.10	
	Living status	Lonely	12.30	0.033 *
		Not lonely	87.70	
	Work status	Yes	57.10	0.150
		No	42.90	
	Age	≤18	11.00	0.006 *
		19–45	70.80	
		≥46	18.20	
	Having special family members to care for	Yes	41.20	0.026 *
		No	58.80	
	Family harmony	Not at all	2.7	0.000 *
		A little	2.7	
		General	13.5	
		More	17.9	
		Very	63.2	
	Indoor space	Very crowded	8	0.000 *
		More	5.6	
		General	12.7	
		A little	11	
		Not at all	62.7	
Lifestyle changes	Dietary habits	Less healthy	22.2	0.000 *
		No change	51.5	
		Healthier	26.3	
	Physical activity	Less than before	69.7	0.000 *
		No change	22.8	
		More than before	7.5	
	Sleep	More irregular	48.9	0.000 *
		No change	37.1	
		More regular	14	
	Screen time	Less than before	71.6	0.000 *
		No change	22.9	
		More than before	5.5	
	Smoking and alcohol consumption	Less than before	7.8	0.009 *
		No change	86.1	
		More than before	6.1	
	Interaction with neighbors	Less frequently	54.9	0.063
		No change	31.8	
		More frequently	13.3	
Risk perception		Lower Group	36.7	0.000 *
		Medium Group	38.6	
		Higher Group	24.7	
Anxiety		Minimal	63.8/(1.25 ± 1.41)	
		Mild	24.2/(6.62 ± 1.2)	
		Moderate and severe	12/(14.61 ± 3.95)	

* The significant of mean difference is at the 0.05 level.

**Table 2 ijerph-19-13379-t002:** Associations between lifestyle changes and anxiety.

Lifestyle Changes	B	*p*-Value	CI	
			Lower	Upper
Dietary habits	−0.165	0.000	−1.527	−0.777
Sleep	−0.093	0.001	−1.012	−0.258
Smoking and alcohol consumption	−0.057	0.023	−1.388	−0.103
Screen time	−0.051	0.05	−0.862	−0.001

**Table 3 ijerph-19-13379-t003:** Associations between lifestyle changes, risk perception, and anxiety.

Anxiety Status ^1^	Variables		Unadjusted (Model 1)		Adjusted (Model 2)	
			OR (95% CI)	*p*-Value	OR (95% CI)	*p*-Value
Mild	Screen time ^2^	Negative changes	1.54 (1.05–2.28)	0.029	1.60 (1.08–2.37)	0.019
	Risk perception ^3^	Lower group			0.69 (0.49–0.91)	0.028
Moderate and severe	Dietary habits ^2^	Negative change	3.94 (2.45–6.37)	0.000	3.94 (2.44–6.38)	0.000
	Sleep ^2^	Negative change	1.72 (1.01–2.92)	0.045	1.70 (1.00–2.90)	0.050
	Smoking and alcohol consumption ^2^	Negative change	2.03 (1.03–4.01)	0.041		
	Risk perception ^3^	Lower group			0.58 (0.35–0.96)	0.003

^1^ taking minimal anxiety status as the reference; ^2^ taking “no change” as the reference; ^3^ taking the “Medium Group” as the reference.

**Table 4 ijerph-19-13379-t004:** Influencing factors of anxiety among risk perception groups.

Groups		Variables	B	*p*-Value	95% CI	
					Lower	Upper
Risk perception	Lower Group	Dietary habits	−0.839	0.004	−1.40	−0.28
		Sleep	−0.630	0.029	−1.20	−0.06
		Family harmony	−1.410	0.001	−2.23	−0.60
		Indoor space	−0.955	0.002	−1.56	−0.35
	Medium Group	Dietary habits	−0.598	0.003	−0.99	−0.21
		Sleep	−0.585	0.011	−1.03	−0.14
		Family harmony	−2.123	0.000	−2.86	−1.39
		Indoor space	−0.670	0.023	−1.25	−0.09
		Work status	−0.411	0.033	−0.79	−0.03
		Having special family members to care for	0.461	0.022	0.07	0.86
	Higher Group	Dietary habits	−1.466	0.000	−2.18	−0.75
		Family harmony	−3.686	0.000	−4.69	−2.68
		Net income	−0.867	0.006	−1.48	−0.26

The regression set the classification variables of sex, living status, work status, and having special family members to care for as virtual variables, and took male, not lonely, not working, and no special family members to care for, respectively, as the reference.

## Data Availability

The data can be obtained by contacting the corresponding author.

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
