# Peer review of "Associations between Lifestyle Changes, Risk Perception and Anxiety during COVID-19 Lockdowns: A Case Study in Xi’an"

_ijerph, 2022, doi:10.3390/ijerph192013379_

Round 1

Reviewer 1 Report

The authors presented an interesting study. A good review of the literature, which includes the opinions of scientists from different countries. Interesting conclusions; however, there are significant shortcomings in the description of the research methodology: there is no transparent and clear presentation of the research methodology in all aspects: from the beginning of collecting information to the end of its processing.

Reviewer 2 Report

The study is interesting. However, there are some shortcomings, warranting revision before acceptance. 

Firstly, the economic loss due to COVID-19 should be presented, to justify the importance of the paper. References and statistics support is needed. 

Secondly, the authors mentioned social support. However, social support is received when users engage with each other on social-media platforms. Thus, the authors are recommended to present the importance of social media contents in driving social support. The following references are useful for the improvement of this paper: 

Cheung, M. L., Leung, W. K., Aw, E. C. X., & Koay, K. Y. (2022). “I follow what you post!”: The role of social media influencers’ content characteristics in consumers' online brand-related activities (COBRAs). Journal of Retailing and Consumer Services66, 102940.

Yu, M., Li, Z., Yu, Z., He, J., & Zhou, J. (2021). Communication related health crisis on social media: a case of COVID-19 outbreak. Current issues in tourism24(19), 2699-2705.

Obi-Ani, N. A., Anikwenze, C., & Isiani, M. C. (2020). Social media and the Covid-19 pandemic: Observations from Nigeria. Cogent arts & humanities7(1), 1799483.

Reviewer 3 Report

I would like to thank you for giving me this opportunity to evaluate this scientific paper, which focuses on a topic of importance: the changes in individuals’ lifestyles, which include dietary habits, physical activity (PA), sleep, screen time, smoking and alcohol consumption, and interaction with neighbors in the context of covid lockdown. The topic is of interest because it focuses on more exceptional Covid lockdown in the later phases of the pandemic.

This manuscript reports new findings and is theoretically based on the current literature. The sample size is adequate.

The subject is very important for research about lifestyle and mental health

- The manuscript is within the journal's scope.

- This study was well designed, executed, and presented.

- Figures and tables are well presented

- The conclusion is consistent with the evidence presented

- The discussion is relevant

- References are up to date and relevant, but the reference are not in the format required for the journal.

References should be described as follows,:

 Journal Articles:
1. Author 1, A.B.; Author 2, C.D. Title of the article. Abbreviated Journal Name Year, Volume, page range.

Author Response

Point 1: References are up to date and relevant, but the reference are not in the format required for the journal. 

Response 1:  We updated the format of some references.  You can find details in attachment.

Reviewer 4 Report

1. The research question is too simplistic. From the content of the introduction, the author's literature review on the relationship between lifestyle and anxiety, and the relationship between risk perception and anxiety is too simple and not comprehensive enough. Then the paper immediately put forward the research purpose, which is 'to comprehensively evaluate the relationship between lifestyle changes, risk perception and anxiety', in which what does 'comprehensive' mean? can a comprehensive analysis afford the novelty of this research?

2. In the empirical cases, only one example from Xi'an in late 2021 and early 2022 was concerned. It has been nearly two years since the outbreak of the epidemic. During this period, the means of epidemic control should have a continuous impact on people's lives. It is recommended that the author further clarify the general significance of this research.

3. For the indicator system, it is constructed 'directly' through other scholars' works, and there is no clear description for the origins of the indicator system. The so-called 'comprehensive' is more like the accumulation of indicators.

4. This paper uses the method of online questionnaire survey. Nevertheless, fewer elderly people and children participate in it. The sample bias is a flaw, and there is a lack of subjective first-hand data of the elderly and children.

5. The discussion of meaning is not deep enough. The discussion of significance in this paper focuses on the inspiration for policy makers, but in the real world, the research objects of this paper can also alleviate negative emotions during the COVID-19 pandemic through self-activation. Further discussions about the significance of this study should be broadened.

Round 2

Reviewer 2 Report

Although the authors have amended the paper, there are some shortcomings. Please refer to the following: 

The authors are recommended to highlight the theoretical and managerial implications in the introduction section. 

In addition, the authors are highly recommended to mention the importance of social media in affecting individuals' behaviors during the pandemic. The following references are highly recommended to be cited in the paper. 

Cheung, M. L., Leung, W. K., Aw, E. C. X., & Koay, K. Y. (2022). “I follow what you post!”: The role of social media influencers’ content characteristics in consumers' online brand-related activities (COBRAs). Journal of Retailing and Consumer Services66, 102940.

Zhao, Y., Cheng, S., Yu, X., & Xu, H. (2020). Chinese public's attention to the COVID-19 epidemic on social media: observational descriptive study. Journal of medical Internet research22(5), e18825.

Reviewer 4 Report

The logical problems of the article that I've addressed in the last round of comments have been basically improved. But, I still reserve the opinion on the attraction of the theme to the potential readers. Let the journal make the decision.

Author Response

Thank you for your review and valuable advices.